# A human-neutral large carnivore? No patterns in the body mass of gray wolves across a gradient of anthropization

Jacopo Cerri[1]*, Carmela Musto[2], Federico M. Stefanini[3], Umberto di Nicola[4], Nicoletta Riganelli[4], Maria C. Fontana[5], Arianna Rossi[5], Chiara Garbarino[5], Giuseppe Merialdi[5], Francesca Ciuti[6], Duccio Berzi[6], Mauro Delogu[2], Marco Apollonio[1]

1 Department of Veterinary Medicine, University of Sassari, Sassari, Italy, 2 Department of Veterinary Medical Sciences, University of Bologna, Bologna, Italy, 3 Dipartimento di Scienze e Politiche Ambientali, Università degli Studi di Milano "La Statale", Milano, Italy, 4 Gran Sasso and Monti della Laga National Park, L'Aquila, Italy, 5 Istituto Zooprofilattico Sperimentale della Lombardia e dell'Emilia-Romagna Bruno Ubertino, Brescia, Italy, 6 Canislupus Italia, Firenze, Italy

☯ These authors contributed equally to this work.
* jacopocerri@gmail.com

**Data Availability Statement:** Data available via Open Science Framework Digital Repository https://osf.io/g2jsv/.

## Abstract

The gray wolf (*Canis lupus*) expanded its distribution in Europe over the last few decades. To better understand the extent to which wolves could re-occupy their historical range, it is important to test if anthropization can affect their fitness-related traits. After having accounted for ecologically relevant confounders, we assessed how anthropization influenced *i*) the growth of wolves during their first year of age (n = 53), *ii*) sexual dimorphism between male and female adult wolves (n = 121), in a sample of individuals that had been found dead in Italy between 1999 and 2021. Wolves in anthropized areas have a smaller overall variation in their body mass, during their first year of age. Because they already have slightly higher body weight at 3–5 months, possibly due to the availability of human-derived food sources. The difference in the body weight of adult females and males slightly increases with anthropization. However, this happens because of an increase in the body mass of males only, possibly due to sex-specific differences in dispersal and/or to "dispersal phenotypes". Anthropization in Italy does not seem to have any clear, nor large, effect on the body mass of wolves. As body mass is in turn linked to important processes, like survival and reproduction, our findings indicates that wolves could potentially re-occupy most of their historical range in Europe, as anthropized landscapes do not seem to constrain such of an important life-history trait. Wolf management could therefore be needed across vast spatial scales and in anthropized areas prone to social conflicts.

## Introduction

The presence of medium and large carnivores in anthropized environments increased over the last few decades, due to urban sprawl and agricultural development in the Global South [1], and a mix of socio-ecological dynamics and legal protection in the Global North [2–4].

**Funding:** Carmela Musto and Marco Apollonio were partially supported by a research grant funded by the Vienna Science and Technology Fund (Project number: WWTF ESR20-009). The funders had no role in study design, data collection and analysis, decision to publish, or preparation of the manuscript. There was no additional external funding received for this study.

**Competing interests:** The authors declare no competing interest, regarding the study.

Therefore, a growing number of studies explored how well these species can adapt to increased levels of human presence, to improve their conservation planning and reduce the risk of conflicts with human activities.

Anthropization was found to have three main effects over medium and large carnivores. In some cases, anthropized environments are sub-optimal, due to decreased prey availability [5] the influence of human activity and artificial nightlight on foraging [6,7], persistent human disturbance [8], the impact of infrastructures on population connectivity and mortality [9,10], disease transmission and competition with domestic dogs [11] and the risk of accidental intoxication [12]. These dynamics can in turn raise metabolic stress [13], and limit reproduction and survival, thus creating source-sink dynamics with undisturbed areas [14].

On other occasions, carnivores prosper in anthropized environments, attaining higher body sizes [15–17] and densities [18] than those reported for environments with no human presence. Mostly because of reduced competition [19] and the exploitation of alternative food sources [20,21].

Finally, on some other cases, effects are non-linear: moderate levels of anthropization seem to be advantageous [22], or detrimental [13], compared to natural environments, but these effects reverse as anthropization increases.

The expansion of the gray wolf (*Canis lupus*) in Italy and Europe calls for further research to better forecast its future trajectory. Between early 1990s and mid 2010s wolves recolonized marginal areas in Europe, due to increased forest cover and rural abandonment [3]. In Italy, wolves were legally protected since 1976 and benefited from recovering forests and ungulates for decades, progressively saturating undisturbed habitat patches [23] and then further expanding into increasingly anthropized ecosystems, recovering most of their historical range. Nowadays wolves occur in peri-urban areas and even the Po plain, one the areas in Europe with the highest human density (above 650 inhabitants/Km$^2$).

This dynamic in Italy was probably facilitated by the capacity of local wolf population to cope with human presence, as well as to use food waste as a substitute for large ungulates, a prey base that had been depleted for decades [24], and possibly also by the availability of feral and stray dogs for reproduction [25]. These conditions were certainly different at some other areas in Europe, where forest recovery was often less marked than in Italy [26], in turn hampering connectivity between wolf populations [27]. And where wolves were less adapted to human presence, by having been segregated in areas with low disturbance and a stable prey base like in Poland [28,29], Estonia [30], Sweden [31] or Finland [32].

Provided that this heterogeneity in long-term selective pressures might have turned into a different capacity of exploiting human-dominated landscapes, the situation between Italy and other countries from Southern, Central and Eastern Europe, some of whom have indeed been colonized by individuals from the Italian peninsula (e.g., France) [27], seems to be temporally lagged, but following a rather similar trajectory. Wolves are progressively colonizing human-dominated landscapes, initially with single dispersing individuals or couples [33–35], and then either with source-sink dynamics [36,37] and or with well-established packs [38,39] a process that in Italy has been observed 20–15 years ago. Therefore, understanding the suitability of anthropized areas for wolves in Italy, a country where wolf colonization is at a more advanced stage, could be pivotal to evaluate the extent to which the species could re-occupy its historical range in Europe and to forecast the spatial scale of future mitigation measures or zonation policies [40].

Considered the ecology and behavior of the gray wolf, all the three scenarios are equally plausible. In the first one, anthropized areas could be sub-optimal, because replacing large ungulates with smaller prey [41] might raise the energetic costs of foraging, given limited nutritional benefits and because wolves may suffer from disease transmission from domestic

dogs [11]. Alternatively, anthropized areas could be favorable for wolves, which could exploit large amounts of food waste [42]. Finally, wolves could show non-linear response to anthropization, as areas with intermediate level of anthropization could still offer large ungulates together with domestic animals and food waste at the same time.

Considered the ecology of gray wolves, anthropization is likely to act through two mechanisms. In the first year of age, anthropization acts on early-life conditions, like for most mammals [43] affecting growth and in turn survival and the capacity to disperse and reproduce of young wolves. While during their first month wolf pups could be fed with food waste, the role of conventional prey become progressively more important as individuals approach adulthood and hunting strategies are developed [44]. Thus, the extent to which a certain environment is optimal for pups is measured by the strength of the temporal growth in body mass, net the effect of total body length. If the environment provided more food sources, young wolves would grow faster and the linear interaction would therefore be positive, reinforcing the effect of age in days over body mass (positive interaction). On the other hand, if environments were sub-optimal, young wolves would grow slower and the effect of age in days over body mass would be weaker (negative interaction).

From the second year of age, when wolves complete their growth [45,46], anthropization is likely to act through sexual dimorphism. Wolves are moderately dimorphic carnivores, with males weighting around 20% more than females [47]. In wolves males are believed to be larger than females both because they engage more often in intergroup aggressions [48], and because they provide prey to lactating females and pups [49]. In facts, a larger body size is deemed to improve either the hunting proficiency of male wolves [50] and their capacity to kill large prey [51,52]. Considered that in anthropized areas adult wolves could exploit human food waste, not relying uniquely on large prey, we predicted the size of both sexes to increase. However, due to reduced selective pressures for larger body size needed to prey upon large ungulates, we expected the size of adult males to increase less than that of females, leading to a decrease in sexual dimorphism. This study is a first attempt to address these two effects of anthropization, across a gradient of urbanized areas in central Italy and a time span of 22 years.

## Methods

### Study area

The study area encompasses the Emilia-Romagna region, the northern provinces of the Tuscany region, and the Gran Sasso and Monti della Laga National Park, in the Abruzzo region, in Italy (Fig 1).

In Emilia-Romagna and Tuscany, two contiguous regions, a wolf population of at least 110 packs was estimated between 2012 and 2016 [53,54]. In the 90's wolves were dived in two distinct sub-populations, one in the Apennine ridge and one in coastal and hilly part of central-southern Tuscany and Latium, which subsequently merged as the species expanded its distribution around 2013 [55]. In the Gran Sasso and Monti della Laga National Park available estimates indicate a population of 11–14 packs [56]. Both areas suffered from wolf-dog hybridization, conflicts with livestock and illegal wolf killing [53,57].

The landscape includes a variety of different ecosystems, ranging from coastal areas characterized by Mediterranean maquis to temperate broad-leaved forests and sub-alpine grasslands in the Apennines. In the study area, the human exodus from marginal rural areas at higher elevations [58], produced a strong gradient of anthropization (Fig 1). The portion of the study area in the Emilia-Romagna and Tuscany regions hosts a population of 8.2 million people, across 45,438 km$^2$ (180 inhabitants/km$^2$), while the Gran Sasso and Monti della Laga park spans across 1500 km$^2$ with a population of 138,669 people (92.5 inhabitants/km$^2$).

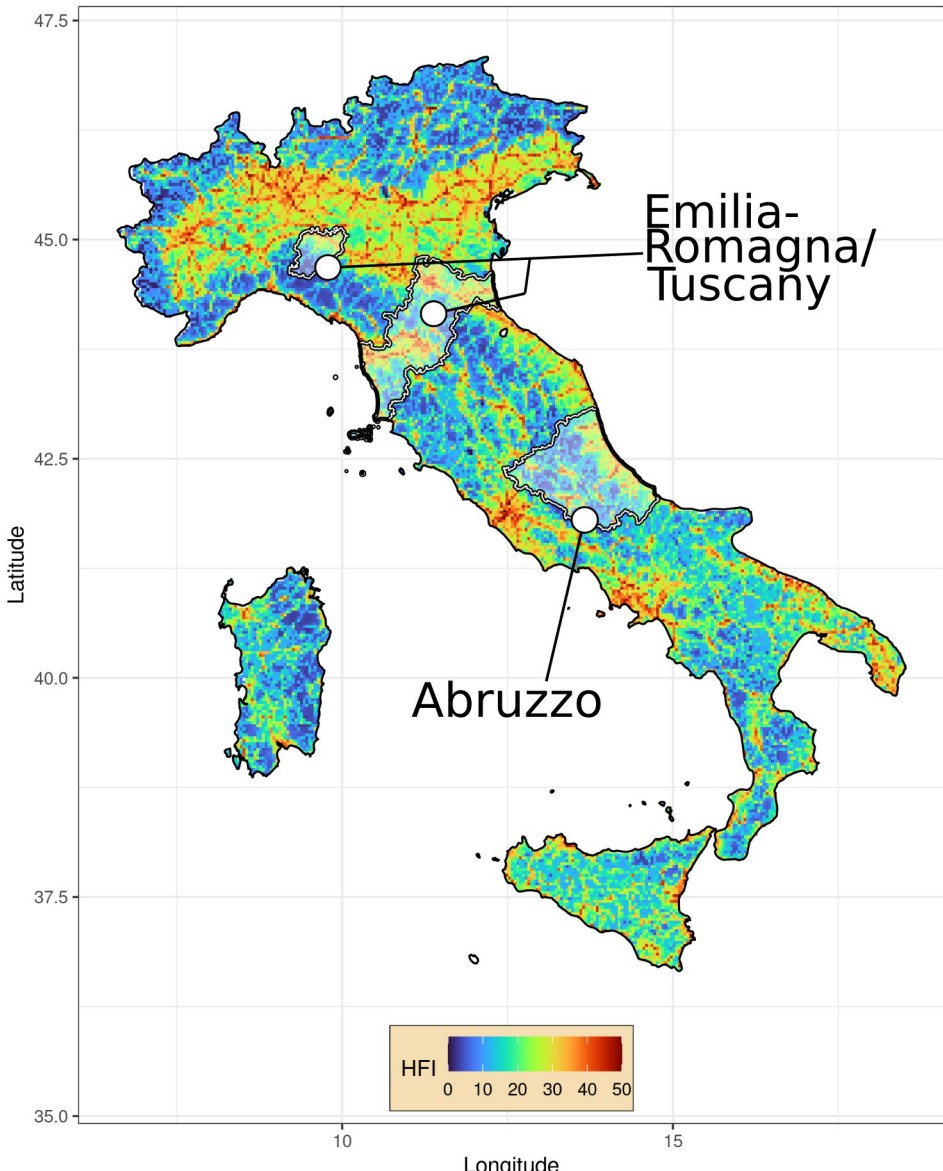

**Fig 1. Map of the study area, representing the human footprint index and provinces covered by data collection (highlighted) in Italy.** The Human Footprint Index was available under a CCBY 4.0 license from Venter et al. (2016, see the references) [76].

The trophic niche of wolves in the study area remained quite broad over the last three decades, being centered on wild/domestic ungulates [59–66] which are available throughout the region, as Central Italy hosts among the highest densities of wild ungulates in Europe [24], and key prey such as roe deer (*Capreolus capreolus*) and wild boar (*Sus scrofa*) are abundant even in croplands and peri-urban areas [67]. However, urban areas are also rich in human food waste. In Italy, each citizen produces between 46 and 127 kg of food waste on each year [68], and waste collection is characterized by periodical inefficiencies and long collection times, with waste sometimes piling up at garbage bins for days and thus becoming available to wildlife (https://www.euronews.com/my-europe/2022/05/18/how-rome-s-rubbish-problem-is-attracting-wild-boar-into-the-italian-capital).

## Collection of dead wolves and laboratory analyses

Our dataset included 107 wolves from the Emilia-Romagna and Tuscany region, as well as 68 wolves from the Abruzzo region. All these animals were dead and recovered by local authorities between 1999 and 2021. The proportion of males, the age of recovered wolves, the season when they were found, and the number of roadkill were similar between Abruzzo and the Emilia-Romagna/Tuscany region (S1 to S4 Figs). A complete overview about our sample is available in Table 1.

The age of each animal was estimated based on dental development, body size and weight [69], dividing individuals between 1–12 months, 13–24 months or older. Until 24 months, individuals were aged by assuming they were born on the 1st of May [70]. We also recorded

**Table 1. Sample overview.** Main attributes of the wolves that were analyzed within this study.

| Age class | | |
|---|---|---|
| | Females | Males |
| First year of age | 25 | 30 |
| Second year of age | 25 | 39 |
| Third year of age, or higher | 23 | 33 |
| Causes of mortality | | |
| | Females | Males |
| Collisions with trains or vehicles | 50 | 64 |
| Persecution | 8 | 11 |
| Natural injuries | 11 | 16 |
| Pathologies | 3 | 10 |
| Unspecified | 1 | 1 |
| Year | | |
| 1999 | 0 | 1 |
| 2000 | 0 | 0 |
| 2001 | 2 | 2 |
| 2002 | 0 | 1 |
| 2003 | 0 | 0 |
| 2004 | 2 | 3 |
| 2005 | 1 | 1 |
| 2006 | 0 | 0 |
| 2007 | 2 | 2 |
| 2008 | 0 | 1 |
| 2009 | 4 | 4 |
| 2010 | 4 | 4 |
| 2011 | 3 | 8 |
| 2012 | 3 | 5 |
| 2013 | 3 | 4 |
| 2014 | 4 | 6 |
| 2015 | 2 | 6 |
| 2016 | 3 | 3 |
| 2017 | 6 | 11 |
| 2018 | 12 | 9 |
| 2019 | 6 | 12 |
| 2020 | 12 | 13 |
| 2021 | 4 | 6 |

total length (from the nose to the junction of the tail), the length of the tarsus and tail, the height of the ear, and chest and neck circumferences.

Out of our sample of 175 wolves, 11 of them had not been genetically tested for hybridization with domestic dogs. Among tested individuals, only 8 wolves were recent hybrids.

As the research did not deal with live animals, but instead with animals that had already been found dead, and then subjected to necropsy, it was not necessary to obtain the approval from the animal research ethics committee of the University of Bologna. Moreover, as data were collected by local authorities across the study area, often on public land, no permission to access the study site was needed.

### Individual genotyping and taxonomy of individual

Genetic investigation on carcasses were conducted by Istituto Superiore per la Protezione e la Ricerca Ambientale—ISPRA. A portion of lingual muscular tissue was taken and stored in 95% ethanol to genetically determine the species of the examined canids and to detect a possible presence of genetic hybridization signatures [71,72].

### Statistical analyses and hypotheses

Both for growth and dimorphish, the parameter of interest, which we then used as a response variable in statistical modeling, was the body mass of individuals. Body mass is strongly related to body condition, which, in mammals, is often measured as the ratio between body mass and total body length [73,74]. We rather modeled the contribution of body length to body mass through a linear predictor, as a covariate-based approach ensured a higher level of flexibility compared to a ratio between the two measures, for example by allowing for non-linear relationships or heteroskedasticity [75].

Body mass was measured as the weight of dead individuals that had been recovered, in kg. Urbanization was measured by means of the Human Footprint Index (hereinafter HFI), obtained by combining multiple layers about man-made structures from satellite, at the resolution of 1 km [76]. The median HFI was calculated in a buffer with a radius of 6 km around the point. This size corresponded to an area of approx. 113 km$^2$ around the point, reflecting the most recent estimates for the home range of the species reported in Italy [77,78].

We controlled for candidate confounding variables, through the so-called "back door criterion" [79]. A complete overview of candidate confounders, acting through some unobserved mediating variables, is shown in the (S5 Fig), in the form of a Directed Acyclic Graph.

We also controlled for some temporal variables that characterized sampling. The day of the year could have affected the level of urbanization of recovered wolves, because human presence in natural environments, and the probability of recovering wolf carcasses, is higher during summer or during the hunting season, which lasts from autumn and early winter. In adult wolves, these seasons could also be characterized by a higher availability of prey, such as young ungulates, compared to winter and spring. Moreover, our data collection covered 22 years and thus we controlled for the year on which each wolf was found. As the wolf population steadily expanded its distribution in Italy, wolves were forced to disperse more and more in urbanized areas [80], while at the same time they could also have increased their average body condition due to the increased abundance of prey species, such as large ungulates. Moreover, as our data had been collected on two separate geographical blocks, corresponding to the Emilia-Romagna/Tuscany and the Abruzzo region, we controlled for this spatial heterogeneity with a dichotomous covariate.

As we found that wolves in the Emilia-Romagna/Tuscany area had a more heterogeneous total body length, we controlled for this variable also on the conditioned variance of total body mass [81].

Models were fitted through the "brms" R package [82]. After a preliminary exploration of body mass distribution, the response variable was modeled as a Student's-t distribution which, for wolves of one year of age, was truncated at zero. Explanatory variables were standardized and centered, before being included in the model. For each parameter, we selected a moderately informative prior distribution, corresponding to a Normal distribution with mean equal to zero and a variance of one [83]. Models had 5000 MCMC iterations and a burn-in of 1000 iterations. Model selection was based on a backward approach, starting by the most complex model with a spline term and removing one term per time, then comparing nested models by means of leave-one-out cross validation [84].

We performed two types of sensitivity analyses. First, we tested for "collapsibility", or the extent to which our interaction terms of interest were susceptible to the removal of confounders, that were deemed redundant by leave-one-out cross. Ideally, the removal of unnecessary confounders should not have changed the posterior distribution of interaction terms that we were interested in. Then, once we identified the best candidate model, we performed a sensitivity analysis, by refitting models with the median HFI, calculated on buffers with a radius between 4 and 16 km. This practice created circles with an area between 12 and 800 km$^2$, which exceeded the whole spectrum of values reported for the core area and the home range of the species in Italy, which attains a maximum of approx. 400 km$^2$ [77].

## Results

The best candidate model predicting wolf growth during their first year explained 79.0% of total variability in the body mass. The year when wolves were found was the only confounding variable that was retained. The body mass of young wolves increased throughout their first year of age but, net the effect of body length, the magnitude of this change was rather mild. Moreover, the growth in body mass became further milder as anthropization increased (Table 2, Fig 2).

The best candidate model predicting differences in body mass between adult male and female wolves explained 57.8% of total variability in the response. The best candidate model

**Table 2. Outputs of the best candidate model for body mass of wolves in their first year of age, and for adult wolves.** Anthropization is calculated on a buffer with a 6km radius around the points where animals had been found.

| Body mass in wolves of 1 year of age | | | |
|---|---|---|---|
| | Estimate | S.E. | 95% Credibility Interval |
| Intercept | 0.03 | 0.05 | (-0.07)–(0.14) |
| Age in days | 0.06 | 0.06 | (-0.06)–(0.18) |
| Anthropization | 0.03 | 0.06 | (-0.08)–(0.13) |
| Year when animals were found | -0.12 | 0.06 | (-0.23)–(-0.01) |
| Total body length | 0.75 | 0.07 | (0.61)–(0.88) |
| Age in days * Anthropization | -0.09 | 0.06 | (-0.20)–(0.03) |
| Body mass in adult wolves. | | | |
| | Estimate | S.E. | 95% Credibility Interval |
| Intercept | -0.36 | 0.09 | (-0.53)–(-0.19) |
| Sex (Male) | 0.58 | 0.12 | (0.36)–(0.80) |
| Anthropization | 0.08 | 0.10 | (-0.13)–(0.28) |
| Total body length | 0.59 | 0.06 | (0.48)–(0.71) |
| Sex (Male) * Anthropization | 0.15 | 0.13 | (-0.12)–(0.40) |

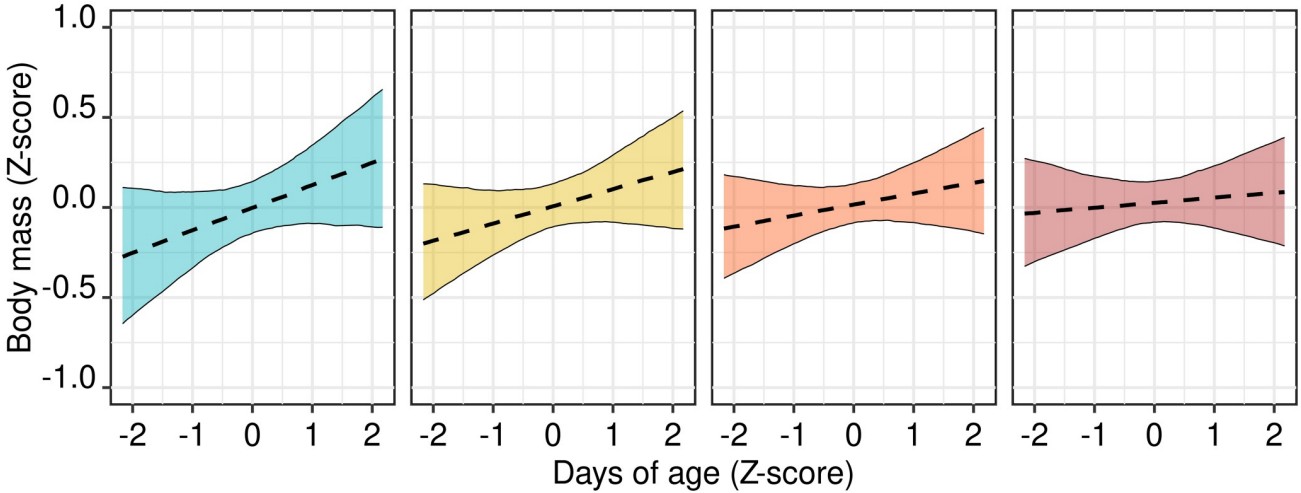

**Fig 2. Interactive effect of anthropization and the age in days of recovered wolves of 1 year of age (n = 53).** Plots correspond to the first (a), second (b), third (c) and fourth (d) quartiles of the distribution of median Human Footprint Index, calculated in a buffer with a 6-km radius around the point where animals were found. Plots (a) to (d) therefore corresponds to increasingly urbanized areas. Variables are standardized and centered.

retained the area where individuals had been found as a confounder, and as a predictor of variability in body mass. The body mass of male wolves showed a mild increase, for increasing levels of anthropization, but the body mass of females did not (Table 2, Fig 3). This decreased predictive accuracy, compared to the model for wolves in their first year of age, probably depended upon the impossibility of correctly aging individuals older than three years, and thus to account for age-related variability in their body mass, which increases until 6–8 years of age [85].

The removal of redundant confounding variables did not change the interactive effect between anthropization and the age of wolves during their first year, nor the interaction between anthropization and the sexual dimorphism in weight among adult wolves (S7 and S8 Figs, S1 and S2 Tables). Moreover, findings from best candidate models did not change, when

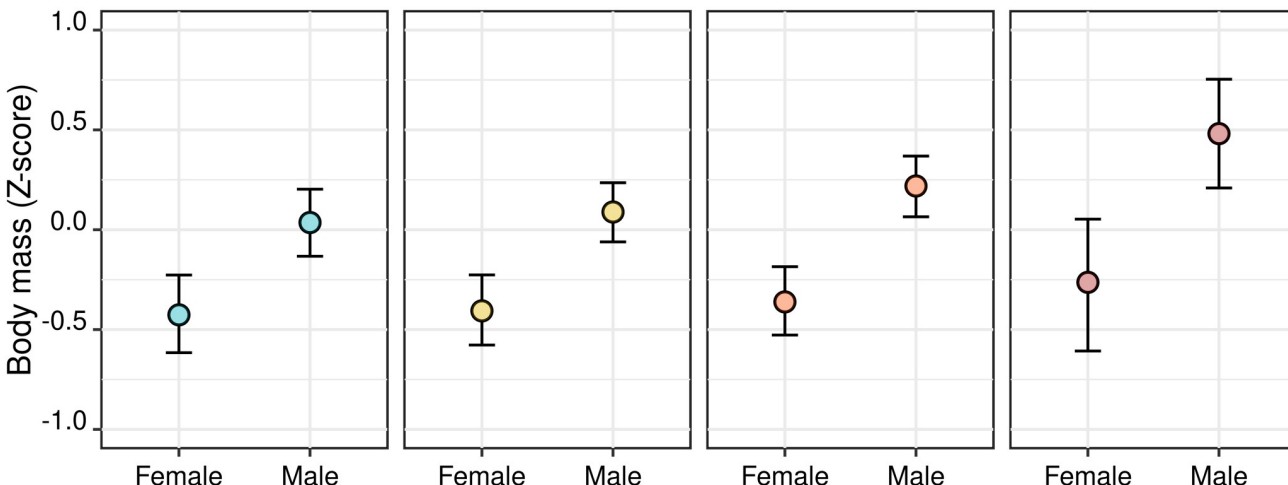

**Fig 3. Interaction between anthropization and the sex of recovered adult wolves (n = 121).** Plots correspond to the first (a), second (b), third (c) and fourth (d) quartiles of the distribution of median Human Footprint Index, calculated in a buffer with a 6-km radius around the point where animals were found. Plots (a) to (d) therefore corresponds to increasingly urbanized areas. Variables are standardized and centered.

calculating anthropization on buffers of different size (Animation S1, S2, available at https://osf.io/g2jsv/). In both models, the analyses of model residuals indicated linear relationships and the semivariogram did not highlight any residual isotropic spatial correlation (S8 to S13 and S14 to S17 Figs). The analysis of posterior predictive checks indicated that the Student's-t distribution was a suitable approximation for the response variable, and the posterior distribution of model parameters indicated model convergence.

## Discussion

To the best of our knowledge, this research was among the first ones [22] assessing the impact of anthropization over the body condition of a large carnivore, the gray wolf, which is occupying increasingly anthropized landscapes in Italy and Europe. While other studies proved that the gray wolf can exploit foods resources characterizing anthropized environments [41,42], we assessed if these environments can affect the body condition of individuals. Our analyses focused on responses measured at the landscape scale, and we did not quantify resource selection or the effect of time-varying resource availability.

This question is non-trivial, because the increased availability of alternative food sources, characterizing anthropized environments, does not automatically translate into an increased body condition of large carnivores. While food waste is abundant and rather predictable in space, thus decreasing foraging costs [86], its nutritional quality might be suboptimal, as it includes a high proportion of carbohydrates which are not processed by wolves which, compared to domestic dogs, are lacking alpha-amylase [87] and have different gut microbiotas [88]. Moreover, foraging in urban environments is influenced by human disturbance. For example, multiple studies reported negative physiological responses of birds and mammals to human disturbance, like increased oxidative stress or inflammation [89]. As for large carnivores, available evidence indicates that avoidance of humans leads pumas (*Puma concolor*) living in anthropized landscapes, to engage in energetically expensive, and inefficient, movement behavior [90].

Our findings provide preliminary evidence that wolves, at least in Italy, are capable to cope with anthropized landscapes in a way which is different from what has been reported for other large carnivores: anthropized landscapes seem not to have any clear effect on the body mass of individuals, an important life-history trait. Or, most likely, anthropized landscapes do not show any clear tradeoff between negative and positive effects.

As we adopted a Bayesian framework for inference, we did not use p-values [91]. Instead, our considerations about effects being significant were based on their magnitude, which in our case largely overlapped with zero.

The body mass of young wolves in our sample grew throughout their first year of age, net the effect of an increase in total body length. This reflected an increase in their muscular mass and fat, two components of body condition. However, this growth had only a very mild interaction with anthropization (Fig 2). More anthropized areas had young wolves experiencing a smaller variation in their body mass during their first year, but which have higher body masses at 3–5 months of age. This suggests a positive influence of human derived food sources in the first months of life, i.e. when pup rising is entirely dependent from adults provisioning. This pattern does not provide any clear evidence on whether anthropized environments are optimal or suboptimal for young wolves in their first year of life. Perhaps anthropized environments can provide an abundance of food resources which could be exploited by young wolves during their first weeks, but less prey such as large ungulates which may be better to sustain individuals at 10–12 months. These differences could also be explained with a different size of wolf packs. Anecdotical evidence indicates that anthropized environments in Italy might have smaller packs often represented by a pair with pups. This strategy can be favorable in areas

with fewer large prey [92] and a rich disposal of small food items, represented by garbage and domestic pets, but might not pay out well in most of our study area, characterized by high densities of ungulates [24].

The effect of anthropization over sexual dimorphism in adult wolves is also partially unclear. While we expected dimorphism to decrease, we instead found a mild increase, as areas became more anthropized, in the body mass of adult males (Fig 3). Most interestingly, the increase in weight involved only male wolves, while females did not increase in body mass at all, despite urbanized areas should be favorable to both sexes. Even if in large mammals it was shown that food shortage/abundance may constrain/favor the growth of males and females differentially (e.g., wild ungulates) [93], the lack of any increase in the weight of females, under potentially favorable conditions, opens the way to n another explanation. In wolves, dispersal seems to be more common for males [53,94] and in mammals, individuals who disperse usually have a larger size, known as "dispersal phenotype" [95]. Considered the progressive saturation of undisturbed habitats by wolf packs [23,80], the pool of dispersing individuals, could have been constituted preferably by large males, which would have dispersed towards more anthropized areas and they could have entered our sample.

Assessing the effect of anthropization on the body mass of wolves is also an urgent question. Body condition is strongly associated with reproduction [96] and survival in mammals, two demographic parameters that are paramount for the long-term viability of populations. Considering the rapid expansion of the gray wolf in Europe, if anthropized environments do not have any effect on the body mass of individuals, as suggested by our study, this could mean that soon wolves could colonize, reproduce, and survive in a significant portion of their historical range in Europe despite a consistent recent urbanization. Thus, policies for co-existing with them, such as zonation or mitigation measures [40], will be needed across vast spatial scales and their implementation might generate a widespread social debate, as it might go beyond rural areas [97].

Considered the potential impacts of a widespread wolf presence in anthropized landscapes of Europe, our findings urgently call for replication studies, addressing two main points. First, studies should replicate our analyses in other geographical areas. Wild ungulates are core prey for wolves and our study covers some of the areas in Europe with the highest densities of ungulates, whose populations increased over the last two decades and which became widespread even in urbanized settings [98]. Other European countries faced a decrease in the wild boar, a key prey, due to the African Swine Fever [99], have high numbers of unprotected livestock and a different amount of waste in the environment. Thus, in these areas, the impact of anthropization over the body mass of wolves can be different.

At the same time, we also emphasize the need for studies based on much larger samples. While we found no significant effect of urbanization on the growth and sexual dimorphism of wolves, considered our sample size we cannot rule out that an effect with a low magnitude exists statistical power increases with sample size and nuanced interactions can be reliably quantified only by analyzing thousands of individuals. In this study we did not perform power analysis, because we had no prior knowledge about effect size and because we aimed to provide only preliminary evidence. However, if other studies will adopt similar sample sizes, there will be no significant advancement in terms of statistical power and it will be impossible to capture small environmental effects, which could nevertheless be potentially important in the long term. Moreover, large-scale data sharing initiatives are needed also to address two other limitations of this study. First, the gray wolf is distributed across the entire Palearctic, inhabiting a variety of different ecosystems, characterized by different prey assemblages, as well as different interactions with humans. It is therefore possible that in these contexts the effect of anthropization over fitness-related traits might differ from our findings, obtained in a Mediterranean

country, being mediated by different behavioral, ecological, and evolutionary mechanisms. As we explained in the Introduction, wolves in Italy had been subjected to higher and more prolonged anthropogenic pressures, than other populations in Europe. Like in the case of other mammals, this could have favored the selection of specific personality traits [100] that subsequently allowed individuals to thrive in human-dominated landscapes [101]. For example, in countries where wolves have always lived in areas with low human densities, individuals consistently seem to avoid human settlements and activities [102,103]. It is also unclear the extent to which hybridization with domestic dogs, which in Italy is high, could have made wolves more capable of tolerating human disturbance or more capable of processing carbohydrates and exploit food waste, as amylase activity is highly variable even among domestic dogs [104]. For example, although no evidence is available for gray wolves and domestic dogs, in North America hybrids between wolves and coyotes (*Canis latrans*), a human-tolerant canid, select more anthropized landscapes than wolves [105]. These findings, altogether with the fact that our recovered wolves did not constitute a randomly selected sample, but perhaps included individuals with some specifically shy personality traits, call for further research at the European scale. Apart from this, although our study focused on body mass, numerous studies has shown that a combination of different morphometric measures can better quantify sexual dimorphism [106]. Although these measures are usually collected from museum specimens, it is not impossible to obtain them from animals that are found dead. As our findings raise serious questions about the potential expansion of wolves in Europe, research groups should pool together their data through collaborative platforms, and improve their collection through harmonized protocols, to address them.

## Supporting information

**S1 Fig. Number of female and male wolves, between Abruzzo and Emilia-Romagna/Tuscany region.** The number of wolves is shown on the y-axis.
(PNG)

**S2 Fig. Distribution of wolves of different age classes, between Abruzzo and Emilia-Romagna/Tuscany region.** The number of recovered wolves is shown on the y-axis.
(PNG)

**S3 Fig. Distribution of recovered wolves, between Abruzzo and Emilia-Romagna/Tuscany region, across the different months of the year.**
(PNG)

**S4 Fig. Distribution of recovered wolves that had been involved in collisions with vehicles, between Abruzzo and Emilia-Romagna/Tuscany region.** The number of recovered wolves is shown on the y-axis.
(PNG)

**S5 Fig. Directed Acyclic Graph (DAG), showing the causal relationship of interest and candidate confounders that were included in model selection (light color) and unobserved mediators (dark color).** Total body length is not shown, as the predictor was included in the model not as a confounder, but to rule out the part of body mass that did not depend upon body condition, but upon differences in the size of animals.
(PNG)

**S6 Fig. Effect of the removal of redundant confounders, over the interaction between anthropization and the age in days of recovered wolves of 1 year of age.** Plots correspond to the first (a), second (b), third (c), and fourth (d) quartiles of the distribution of median

Human Footprint Index, calculated in a buffer with a 6-km radius around the point where animals were found. Columns (a) to (d) therefore corresponds to increasingly urbanized areas, while rows to models selected in S1 Table.
(PNG)

**S7 Fig. Effect of the removal of redundant confounders, over the interaction between anthropization and the weight dimorphism of adult wolves.** Plots correspond to the first (a), second (b), third (c) and fourth (quartiles) of the distribution of median Human Footprint Index, calculated in a buffer with 6-km radius around the point where animals were found. Columns (a) to (d) therefore corresponds to increasingly urbanized areas, while rows to models selected in S2 Table.
(PNG)

**S8 Fig. Residuals versus fitted values from the best candidate model.**
(PNG)

**S9 Fig. Residuals from the best candidate model versus standardized and centered scores of the Human Footprint Index.**
(PNG)

**S10 Fig. Residuals from the best candidate model versus standardized and centered scores of the age in days of recovered wolves.**
(PNG)

**S11 Fig. Residuals from the best candidate model versus standardized and centered scores of the year when wolves were recovered.**
(PNG)

**S12 Fig. Residuals from the best candidate model versus standardized and centered scores of the total body length.**
(PNG)

**S13 Fig. Isotropic semivariogram: Semivariance of model observations in function of distance between sites where wolves were recovered.**
(PNG)

**S14 Fig. Residuals versus fitted values from the best candidate model.**
(PNG)

**S15 Fig. Residuals from the best candidate model versus standardized and centered scores of the Human Footprint Index.**
(PNG)

**S16 Fig. Residuals from the best candidate model versus standardized and centered scores of the total body length.**
(PNG)

**S17 Fig. Isotropic semivariogram: Semivariance of model observations in function of distance between sites where wolves were uncovered.**
(PNG)

**S1 Table. Theoretical expected log-pointwise predictive density (ELPD) and its standard error (S.E.), obtained from leave one-out cross-validation.** Outputs from models about the body mass of wolves in their first year of age (left).
(PDF)

**S2 Table. Theoretical expected log-pointwise predictive density (ELPD) and its standard error (S.E.) from leave one-out cross-validation.** Outputs from models about adult wolves.
(PDF)

## Author Contributions

**Conceptualization:** Jacopo Cerri, Carmela Musto, Umberto di Nicola, Maria C. Fontana, Arianna Rossi, Chiara Garbarino, Francesca Ciuti, Marco Apollonio.

**Data curation:** Carmela Musto, Umberto di Nicola, Maria C. Fontana, Arianna Rossi, Chiara Garbarino, Giuseppe Merialdi, Francesca Ciuti, Duccio Berzi, Mauro Delogu.

**Formal analysis:** Jacopo Cerri, Federico M. Stefanini.

**Funding acquisition:** Carmela Musto, Mauro Delogu.

**Investigation:** Jacopo Cerri, Nicoletta Riganelli, Maria C. Fontana, Arianna Rossi, Giuseppe Merialdi, Francesca Ciuti, Duccio Berzi.

**Methodology:** Jacopo Cerri, Carmela Musto, Federico M. Stefanini, Umberto di Nicola, Nicoletta Riganelli, Maria C. Fontana, Arianna Rossi, Chiara Garbarino, Giuseppe Merialdi, Francesca Ciuti, Mauro Delogu, Marco Apollonio.

**Project administration:** Nicoletta Riganelli, Maria C. Fontana, Marco Apollonio.

**Resources:** Nicoletta Riganelli, Arianna Rossi, Chiara Garbarino, Giuseppe Merialdi, Duccio Berzi, Marco Apollonio.

**Software:** Jacopo Cerri.

**Supervision:** Jacopo Cerri, Carmela Musto, Federico M. Stefanini, Maria C. Fontana, Chiara Garbarino, Francesca Ciuti, Duccio Berzi, Mauro Delogu, Marco Apollonio.

**Validation:** Jacopo Cerri, Francesca Ciuti, Duccio Berzi, Mauro Delogu, Marco Apollonio.

**Visualization:** Jacopo Cerri.

**Writing – original draft:** Jacopo Cerri, Carmela Musto, Federico M. Stefanini, Nicoletta Riganelli, Duccio Berzi, Marco Apollonio.

**Writing – review & editing:** Jacopo Cerri, Carmela Musto, Marco Apollonio.

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
