## [Decision Letter · Decision Letter 0]

16 Nov 2022

PONE-D-22-27536A human-neutral large carnivore? No patterns in the body mass of gray wolves (Canis lupus) across a gradient of anthropizationPLOS ONE

Dear Dr. Cerri,

Thank you for submitting your manuscript to PLOS ONE. After careful consideration, we feel that it has merit but does not fully meet PLOS ONE’s publication criteria as it currently stands. Therefore, we invite you to submit a revised version of the manuscript that addresses the points raised during the review process.

We look forward to receiving your revised manuscript.

Kind regards,

Paulo Corti, Ph.D.

Academic Editor

PLOS ONE

Journal Requirements:

3. Please amend your current ethics statement to address the following concerns:

a) Did participants provide their written or verbal informed consent to participate in this study?

"Carmela Musto and Marco Apollonio were partially supported by a research grant funded by the Vienna Science and Technology Fund (Project number: WWTF ESR20-009)"

Please state what role the funders took in the study.  If the funders had no role, please state: ""The funders had no role in study design, data collection and analysis, decision to publish, or preparation of the manuscript."" If this statement is not correct you must amend it as needed. 

5. Thank you for stating in your Funding Statement: 

"Carmela Musto and Marco Apollonio were partially supported by a research grant funded by the Vienna Science and Technology Fund (Project number: WWTF ESR20-009)"

6. We note that Figure 1 in your submission contain map images which may be copyrighted. All PLOS content is published under the Creative Commons Attribution License (CC BY 4.0), which means that the manuscript, images, and Supporting Information files will be freely available online, and any third party is permitted to access, download, copy, distribute, and use these materials in any way, even commercially, with proper attribution. For these reasons, we cannot publish previously copyrighted maps or satellite images created using proprietary data, such as Google software (Google Maps, Street View, and Earth). For more information, see our copyright guidelines: http://journals.plos.org/plosone/s/licenses-and-copyright.

(1) You may seek permission from the original copyright holder of Figure 1 to publish the content specifically under the CC BY 4.0 license.  

8. In your Data Availability statement, you have not specified where the minimal data set underlying the results described in your manuscript can be found. PLOS defines a study's minimal data set as the underlying data used to reach the conclusions drawn in the manuscript and any additional data required to replicate the reported study findings in their entirety. All PLOS journals require that the minimal data set be made fully available. For more information about our data policy, please see http://journals.plos.org/plosone/s/data-availability.

**Additional Editor Comments:**

Both reviewers did an extensive revision of your manuscript with useful comments to improve it. Pay attention to their suggestions, especially to the organization of the article. Provide with answers of the questions raised by them.

Reviewers' comments:

Reviewer's Responses to Questions

**Comments to the Author**

1. Is the manuscript technically sound, and do the data support the conclusions?

Reviewer #1: Partly

Reviewer #2: Yes

2. Has the statistical analysis been performed appropriately and rigorously? 

Reviewer #1: I Don't Know

Reviewer #2: Yes

3. Have the authors made all data underlying the findings in their manuscript fully available?

Reviewer #1: Yes

Reviewer #2: No

4. Is the manuscript presented in an intelligible fashion and written in standard English?

Reviewer #1: No

Reviewer #2: Yes

5. Review Comments to the Author

Reviewer #1: General comments to the authors

In this manuscript, Cerri and colleagues examine changes in body mass in Italian gray wolves, including possible differences between males and females, in areas affected by varying degrees of anthropization. The authors note the important range expansion that has taken place in recent decades for European wolves and aim to link their results from Italy to the species’ continental range expansion. They discuss the potential influence and cost-benefits of anthropogenic food resources, which are highly relevant issues for wolf evolution and conservation.

The study addresses important topics for wolf and large carnivore conservation and human-wildlife relationships, but I found the structure of the manuscript somewhat difficult to follow. I therefore think more attention toward the organization of the text and the description of the problem statement and aims would greatly improve the manuscript, also considering the journal's wide audience. This includes the order of presentation for certain topics, and the inclusion of additional factors that are noted only briefly or not mentioned. I also recommend reading the revised manuscript carefully with attention to English grammar.

The study is centered on analyses of body mass, and you provide important information in L51-68, including reference to earlier analyses such as Ciucci et al. 2020 on the possible influence of anthropogenic resources. However, it may not be immediately clear to all readers why you are investigating dimorphism in wolves; you note this aspect in L136-140 but provide explanations first in L156 onwards. I also think the addition of broader considerations in evolutionary ecology and behaviour would provide helpful context, including a brief note of the possible influence of wolf-dog hybridization for the ecological role of wolves in your study area and beyond (e.g., Bassi et al. 2017, cited in the study, Galaveri et al. 2017, doi:10.1093/molbev/msx169, Salvatori et al. 2019, https://doi.org/10.1007/s10344-019-1313-3, and Pilot et al. 2021, doi:10.1111/EVA.13257). You mention wolf-dog hybridization in L101, but a brief note on how wolf-dog hybridization may influence selective pressures, also in relation to carbohydrate digestion (L252-255), would add useful context.

The relationship between wolves and wolf body mass in Italy, versus the situation for wolves across the continent, needs further attention. In L74-77 you noted that “understanding the suitability of anthropized areas for wolves in Italy could be pivotal to evaluate the extent to which the species could re-occupy its historical range in Europe ….”. However, a very important cautionary statement on the extent to which the Italian or Mediterranean wolves and wolf habitats may differ from other areas occupied by wolves is noted only at the very end in L322-326.

The habitat, prey, and extent of human modification are quite different in other areas, where the role of larger prey such as moose (e.g., Sand et al. 2017, DOI:10.1371/journal.pone.0168062) is likely to have an important influence on selection in wolves. Such differences suggest that interpretation of wide-scale selective pressures in European wolves based on results from Italy should be done with caution, and I was therefore surprised by your statement in L329-330, although I fully agree with the recommendation for collaboration on further research using standardized methods (L330-332).

For the discussion of the association between body mass and the capacity to (re)colonize, survive and reproduce in various parts of Europe (e.g., L29-31, L297-304), it may also be relevant to briefly note that the smaller golden jackal is now expanding and reproducing in new parts of Europe (e.g., in Estonia; Männil & Ranc 2022, https://doi.org/10.1007/s13364-021-00615-1) and has recently been reported in areas such as northwestern Russia (Rykov et al. 2022, https://doi.org/10.1007/s00300-022-03037-0), showing that the survival of a smaller wolf-like canid is possible in various ecosystems.

Another question I had when reading the text was whether found-dead wolves are representative for wolves in general in Italy and neighbouring regions with similar landscapes, or whether this could have influenced the available samples. Although performed on other canids in different ecosystems, the findings from Sears et al. (2003, Canadian Field-Naturalist 117(4): 591-600, available at https://www.canadianfieldnaturalist.ca/index.php/cfn/article/view/828/828) found canids from landscape types with lower road density, more forest cover and less fragmentation to have more wolf-like morphology and consume larger prey, whereas canids from landscapes with higher road density, less forest cover and more fragmentation were found to be smaller (more coyote-like) and to consume smaller prey species. It is a major effort to sample wolves and you have done considerable work to obtain the existing measures, but given that found-dead wolves seem more likely to be detected in areas with higher human activity, might data from other regional studies that have included live-capture of wolves (e.g., Mancinelli et al. 2019, https://doi.org/10.1111/jzo.12708, Salvatori et al. 2019, noted above) provide at least some comparison to address possible discrepancies?

For model selection, you noted in L195-197 that you used a backward approach and started with the most complex model, then removing one term per time. Was there a particular aim in using this method instead of a forward approach, starting with examination of the effects of single variables?

Specific comments with line numbers

L35, L152, L266 and elsewhere: please pay attention to the difference between “to grow” (“increase in size or substance” and “to grow up” (usually meaning “to reach maturity; become adult”). Given the wording here I think you mean the former. This term is quite important in your text, but its use is not quite clear.

L35-37: This phrase, stating with “Because” appears incomplete, as readers would expect “Because X, therefore Y” and the last part is missing. Overall, I think the abstract needs to be reworded and reorganized to clarify the study background, aims and conclusions (please also see the above comments).

L47: Words that are in the title usually do not need to be included in the keywords and vice versa. Here, I would suggest keeping “Canis lupus” in the key words but not in the title, as there are no other wolf species in Europe, and retain “grey wolves” in the title.

L66: “in some areas”

L75: please remove the comma after “Italy”

L77: Altered legal status and social changes may also have been important here (see e.g., Chapron et al. 2014, cited in the text).

L78: This sentence is not clear; if I understand the meaning correctly I suggest “Considering” at the start, and removing “all the”, as “the” implies that you have already explained what these scenarios are, whereas this information follows below the statement.

L79-80: Do you mean “wolves could not efficiently replace large ungulates as a food source”?

L84: It is not fully clear what “this gap” refers to

L86: “In [or “across”] a time span of 22 years”

L101: “illegal wolf killings”

L104: I think you here mean the exodus of humans?

L110: Suggested “available throughout the region”

L120-132: After the section headline, I recommend starting with L125-132 that explains your sampling and then L120-124 on how you assessed each sample.

L131: Should this procedure, when carried out by humans on wolves (or other animals), not be described as “necropsy”?

L139 and throughout the text: The offspring of wolves are usually called “pups” and not “cubs” (e.g., Barber-Meyer et al. 2021, https://doi.org/10.1016/j.biocon.2021.109145) although you have used “pup/s” in L268 and L275.

L143-145: It is unclear what “rather” means in this phrase, and when you state “ensured a higher level of flexibility”, it is not obvious what this is measured against (another type of predictor?)

L147: Is this your finding, or are you referring to other studies?

L148 and throughout the text: Usually “prey” is used, also in plural.

L161-166: I suggest rewording this section for clarity, starting with L164-166 to first explain that you are expecting increased body mass in both sexes because of anthropogenic food, and then note that you are expecting reduced growth in males because of reduced selective pressures for larger body size, followed by L163-164.

266: I am not sure what is meant with “grew up in a less pronounced way”

L270: Please remove the comma after “Perhaps”

L284: Please check this line after “opens”

L298: “associated with”

L299: “Considering the”

L304: It is not clear what “it” refers to in this phrase. Also, for this section, L299-302 and L302-304 are not well connected, and please see the general comments above.

Reviewer #2: The manuscript deals with an interesting topic. The Authors investigated if and how anthropization can affect fitness-related traits of the gray wolf in Europe. The study is based on good material, results may be used in the population conservation of the wolf.

Nevertheless, I have found some parts in the manuscript which need improvement.

Should handle statistical analyzes and hypotheses separately. A part of it fits better in the Introduction.

Most samples do not come from random sampling (roadkill). I recommend mentioning this.

Instead of Figures S1 and S2, the sample numbers should be presented in a summary table according to age group, sex, year of collection and mortality pattern. The number of samples per year is relatively small.

From Line 119.

It should be mentioned whether there were wolf-dog hybrids among the examined wolves, the seasonality of reproduction may differ from that of wolves.

Based on a study, the authors considered May 1 the date of birth. There can be a difference of up to two months in reproduction and, thus, in the time of pupping. Studies should be mentioned from other seasonally breeding carnivorous species, which estimated the young's age similarly.

In the Introduction and Discussion sections, studies from other social and non-social carnivore species should be mentioned, which analyze the influence of the human environment and the shift in body weight resulting from changes in the food supply.

The Discussion sometimes lacks references or examples for findings (e.g. Lines 249-251, 257, 266-273, 297-299, 321-322).

6. PLOS authors have the option to publish the peer review history of their article (what does this mean?). If published, this will include your full peer review and any attached files.

Reviewer #1: No

Reviewer #2: No

---

## [Author Response · Author response to Decision Letter 0]

9 Dec 2022

#################################################################################Note

################################################################################

#################################################################################Comments

################################################################################

Comment: In your Methods section, please provide additional information regarding the permits you obtained for the work. Please ensure you have included the full name of the authority that approved the field site access and, if no permits were required, a brief statement explaining why.

Reply: as we already explained at lines XXXX, as our research did not deal with live animals, it was not necessary to obtain any particular permit from the ethics committee of the University of Bologna and Experimental Zooprophylactic Institute of Brescia. Necropsies were a regular duty for the Department of Veterinary Medical Sciences and Experimental Zooprophylactic Institute. Moreover, no permission to access field site was needed, as data were collected in sites that were public with no restricted access.

Comment: Please amend your current ethics statement to address the following concerns:

 • Did participants provide their written or verbal informed consent to participate in this study

 • If consent was verbal, please explain i) why written consent was not obtained, ii) how you documented participant consent, and iii) whether the ethics committees/IRB approved this consent procedure.

Reply: as we already explained in the submission, our study dealt with dead wildlife and therefore it was impossible and not needed to ask for informed consent.

Comment: "Carmela Musto and Marco Apollonio were partially supported by a research grant funded by the Vienna Science and Technology Fund (Project number: WWTF ESR20-009)". Please state what role the funders took in the study. If the funders had no role, please state: ""The funders had no role in study design, data collection and analysis, decision to publish, or preparation of the manuscript."" If this statement is not correct you must amend it as needed. Please include this amended Role of Funder statement in your cover letter; we will change the online submission form on your behalf.

Reply: we included this statement in the Cover Letter.

Comment: Thank you for stating in your Funding Statement: "Carmela Musto and Marco Apollonio were partially supported by a research grant funded by the Vienna Science and Technology Fund (Project number: WWTF ESR20-009)". Please provide an amended statement that declares *all* the funding or sources of support (whether external or internal to your organization) received during this study, as detailed online in our guide for authors at http://journals.plos.org/plosone/s/submit-now. Please also include the statement “There was no additional external funding received for this study.” in your updated Funding Statement. Please include your amended Funding Statement within your cover letter. We will change the online submission form on your behalf.

Reply: we included this statement in the Cover Letter.

Comment: We note that Figure 1 in your submission contain map images which may be copyrighted. All PLOS content is published under the Creative Commons Attribution License (CC BY 4.0), which means that the manuscript, images, and Supporting Information files will be freely available online, and any third party is permitted to access, download, copy, distribute, and use these materials in any way, even commercially, with proper attribution. For these reasons, we cannot publish previously copyrighted maps or satellite images created using proprietary data, such as Google software (Google Maps, Street View, and Earth). For more information, see our copyright guidelines: http://journals.plos.org/plosone/s/licenses-and-copyright. We require you to either (1) present written permission from the copyright holder to publish these figures specifically under the CC BY 4.0 license, or (2) remove the figures from your submission:

 1. You may seek permission from the original copyright holder of Figure 1 to publish the content specifically under the CC BY 4.0 license. We recommend that you contact the original copyright holder with the Content Permission Form (http://journals.plos.org/plosone/s/file?id=7c09/content-permission-form.pdf) and the following text: “I request permission for the open-access journal PLOS ONE to publish XXX under the Creative Commons Attribution License (CCAL) CC BY 4.0 (http://creativecommons.org/licenses/by/4.0/). Please be aware that this license allows unrestricted use and distribution, even commercially, by third parties. Please reply and provide explicit written permission to publish XXX under a CC BY license and complete the attached form.” Please upload the completed Content Permission Form or other proof of granted permissions as an ""Other"" file with your submission. In the figure caption of the copyrighted figure, please include the following text: “Reprinted from [ref] under a CC BY license, with permission from [name of publisher], original copyright [original copyright year].”

Reply: Figure 1 has been obtained from https://sedac.ciesin.columbia.edu/data/set/wildareas-v3-2009-human-footprint/docs, which in turn refer to Venter et al., 2016 (https://doi.org/10.1038/ncomms12558), a work which has been licensed under a Creative Commons Attribution 4.0 International License.

Comment: Please include captions for your Supporting Information files at the end of your manuscript, and update any in-text citations to match accordingly. Please see our Supporting Information guidelines for more information: http://journals.plos.org/.

Reply: we harmonized Supporting information according to your guidelines. However, we removed any citation to Supplementary images in the text, as we would like to upload the supplementary information as a single file, due to the high number of figures and tables inside. In the text, we cited two animations, which could then be inspected by downloading the Supplementary files at https://osf.io/g2jsv/

Comment: In your Data Availability statement, you have not specified where the minimal data set underlying the results described in your manuscript can be found. PLOS defines a study's minimal data set as the underlying data used to reach the conclusions drawn in the manuscript and any additional data required to replicate the reported study findings in their entirety. All PLOS journals require that the minimal data set be made fully available. For more information about our data policy, please see http://journals.plos.org/plosone/s/data-availability.

Upon re-submitting your revised manuscript, please upload your study’s minimal underlying data set as either Supporting Information files or to a stable, public repository and include the relevant URLs, DOIs, or accession numbers within your revised cover letter. For a list of acceptable repositories, please see http://journals.plos.org/. Any potentially identifying patient information must be fully anonymized.

Reply: as we explained both in the Data Availability Statement and the manuscript, reproducible data and software code had been made public, months ago, on the Open Science Framework (OSF). OSF is one of the recommended repositories. Data and code are available at: https://osf.io/g2jsv/

Comment: Important: If there are ethical or legal restrictions to sharing your data publicly, please explain these restrictions in detail. Please see our guidelines for more information on what we consider unacceptable restrictions to publicly sharing data: http://journals.plos.org/plosone/s/data-availability#loc-unacceptable-data-access-restrictions. Note that it is not acceptable for the authors to be the sole named individuals responsible for ensuring data access.

Reply: There are not ethical or legal restrictions about data sharing, this is why they have been made public and uploaded on OSF.

Additional Editor Comments

Comment: Both reviewers did an extensive revision of your manuscript with useful comments to improve it. Pay attention to their suggestions, especially to the organization of the article. Provide with answers of the questions raised by them.

Reply: we addressed in detail each comment raised by reviewers. However, please note that some comments from Reviewer#2 were vague and it was not truly possible to address them in detail (e.g., the reviewer generally recommended to cite more studies, when no other study on this topic has been published, to the best of our knowledge).

Comments to the author

Comment: Is the manuscript technically sound, and do the data support the conclusions? - The manuscript must describe a technically sound piece of scientific research with data that supports the conclusions. Experiments must have been conducted rigorously, with appropriate controls, replication, and sample sizes. The conclusions must be drawn appropriately based on the data presented.

 Comment: Reviewer #1: Partly

 Comment: Reviewer #2: Yes

Comment: Has the statistical analysis been performed appropriately and rigorously?

 Comment: Reviewer #1: I don’t know

 Comment: Reviewer #2: Yes

Comment: Have the authors made all data underlying the findings in their manuscript fully available? - The PLOS Data policy requires authors to make all data underlying the findings described in their manuscript fully available without restriction, with rare exception (please refer to the Data Availability Statement in the manuscript PDF file). The data should be provided as part of the manuscript or its supporting information, or deposited to a public repository. For example, in addition to summary statistics, the data points behind means, medians and variance measures should be available. If there are restrictions on publicly sharing data—e.g. participant privacy or use of data from a third party—those must be specified.

 Comment: Reviewer #1: I don’t know

 Comment: Reviewer #2: Yes

Reply: as we explained both in the Data Availability Statement and in the manuscript, reproducible data and software code have been made available, before the submission to PLoS One, on the Open Science Framework, a recommended repository. They can be downloaded at the following link: https://osf.io/g2jsv/

Reviewer #1

Comment: In this manuscript, Cerri and colleagues examine changes in body mass in Italian gray wolves, including possible differences between males and females, in areas affected by varying degrees of anthropization. The authors note the important range expansion that has taken place in recent decades for European wolves and aim to link their results from Italy to the species’ continental range expansion. They discuss the potential influence and cost-benefits of anthropogenic food resources, which are highly relevant issues for wolf evolution and conservation.

The study addresses important topics for wolf and large carnivore conservation and human-wildlife relationships, but I found the structure of the manuscript somewhat difficult to follow. I therefore think more attention toward the organization of the text and the description of the problem statement and aims would greatly improve the manuscript, also considering the journal's wide audience. This includes the order of presentation for certain topics, and the inclusion of additional factors that are noted only briefly or not mentioned. I also recommend reading the revised manuscript carefully with attention to English grammar.

Reply: thanks for your feedback. We re-arranged the structure of the manuscript and addressed your comments in detail.

Comment: The study is centered on analyses of body mass, and you provide important information in L51-68, including reference to earlier analyses such as Ciucci et al. 2020 on the possible influence of anthropogenic resources. However, it may not be immediately clear to all readers why you are investigating dimorphism in wolves; you note this aspect in L136-140 but provide explanations first in L156 onwards.

Reply: we now explained why we tested growth and dimorphism at the end of the Introduction, and we hope that the manuscript would now be clearer. See lines XXXX.

Comment: I also think the addition of broader considerations in evolutionary ecology and behaviour would provide helpful context, including a brief note of the possible influence of wolf-dog hybridization for the ecological role of wolves in your study area and beyond (e.g., Bassi et al. 2017, cited in the study, Galaveri et al. 2017, doi:10.1093/molbev/msx169, Salvatori et al. 2019, https://doi.org/10.1007/s10344-019-1313-3, and Pilot et al. 2021, doi:10.1111/EVA.13257). You mention wolf-dog hybridization in L101, but a brief note on how wolf-dog hybridization may influence selective pressures, also in relation to carbohydrate digestion (L252-255), would add useful context.

Reply: your suggestion is precious. Hybridization with domestic dogs in Italy is widespread, and it has been occurring for decades. Moreover, dogs are more tolerant to human presence, and capable of processing carbohydrates, due to domestication-induced changes. It is therefore highly plausible that domestication could have induced selective pressures that helped wolves from the Italian population to expand and thrive in human-dominated landscapes. The problem lies in the fact that this is a speculation, because no study explored the consequences of hybridization on wolf behavior, nor on critical ecological parameters, such as survival. However, there are examples from North America, showing that wolf/coyote hybrids indeed select different habitats from wolves, being more tolerant towards human disturbance, and more effective at segregating during critical periods of the year (https://doi.org/10.1002/ecs2.3320). Thus, we preferred to mention the potential effect of hybridization, but then to cite the case of wolf/coyote hybrids, for which concrete evidence of different behavior from wolves exists. See lines XXX.

Comment: The relationship between wolves and wolf body mass in Italy, versus the situation for wolves across the continent, needs further attention. In L74-77 you noted that “understanding the suitability of anthropized areas for wolves in Italy could be pivotal to evaluate the extent to which the species could re-occupy its historical range in Europe ….”. However, a very important cautionary statement on the extent to which the Italian or Mediterranean wolves and wolf habitats may differ from other areas occupied by wolves is noted only at the very end in L322-326.

The habitat, prey, and extent of human modification are quite different in other areas, where the role of larger prey such as moose (e.g., Sand et al. 2017, DOI:10.1371/journal.pone.0168062) is likely to have an important influence on selection in wolves. Such differences suggest that interpretation of wide-scale selective pressures in European wolves based on results from Italy should be done with caution, and I was therefore surprised by your statement in L329-330, although I fully agree with the recommendation for collaboration on further research using standardized methods (L330-332).

Reply: we agree with the need of putting our case study into perspective, also by emphasizing long-term differences between wolves from Italy and other European populations. Wolves in Italy faced a long-term selective pressure from humans, coupled with the almost complete depletion of their natural prey basis. A situation that at other areas of Europe did not occur, as remaining wolf populations were confined to less disturbed areas with a higher availability of large ungulates. Nevertheless, there are two points that must to be mentioned. First, in some European countries, like France or Switzerland, wolf colonization is being carried out by individuals from the Italian peninsula. So, it is carried out by those individuals deriving from populations that had been experiencing long-term human pressures. Moreover, even if some populations were subjected to different selective pressures, their colonization of human-dominated landscapes is being rather similar, although temporally lagged, to what was observed in Italy. However, it is true that these differences might be important. We now introduced them in the Introduction (lines XXXX) and the Discussion section (lines XXXX).

Comment: For the discussion of the association between body mass and the capacity to (re)colonize, survive and reproduce in various parts of Europe (e.g., L29-31, L297-304), it may also be relevant to briefly note that the smaller golden jackal is now expanding and reproducing in new parts of Europe (e.g., in Estonia; Männil & Ranc 2022, https://doi.org/10.1007/) and has recently been reported in areas such as northwestern Russia (Rykov et al. 2022, https://doi.org/10.1007/), showing that the survival of a smaller wolf-like canid is possible in various ecosystems.

Reply: we are aware about the ongoing, rapid, expansion of the golden jackal in Europe. Nevertheless, this point has limited connections with our argument. We mention body mass in order to find a measure of body condition in favorable or unfavorable condition with the aim of exploring the impact of urbanization on the wolf. If this impact is absent, we conclude that wolves can adapt to these new environments. In so doing we are not holding any comparison between canids of different size. Conversely, it is interesting a variation in the body mass of wolves, and thus in dimorphism and growth, as it might provide us with unique insights about how wolves are responding to human presence in an area being sexual dimorphis in many mammals a function of individuals welfare. It would be interesting to cite studies comparing body mass variations in golden jackals, and linking them to landscape characteristics, but we are not aware of their existence.

Comment: Another question I had when reading the text was whether found-dead wolves are representative for wolves in general in Italy and neighbouring regions with similar landscapes, or whether this could have influenced the available samples. Although performed on other canids in different ecosystems, the findings from Sears et al. (2003, Canadian Field-Naturalist 117(4): 591-600, available at https://www.canadianfieldnaturalist.ca/index.php/cfn/article/view/828/828) found canids from landscape types with lower road density, more forest cover and less fragmentation to have more wolf-like morphology and consume larger prey, whereas canids from landscapes with higher road density, less forest cover and more fragmentation were found to be smaller (more coyote-like) and to consume smaller prey species. It is a major effort to sample wolves and you have done considerable work to obtain the existing measures, but given that found-dead wolves seem more likely to be detected in areas with higher human activity, might data from other regional studies that have included live-capture of wolves (e.g., Mancinelli et al. 2019, https://doi.org/10.1111/jzo.12708, Salvatori et al. 2019, noted above) provide at least some comparison to address possible discrepancies?

Reply: honestly, we wondered about the same issue for a long time, but in our opinion there are two considerations that convinced us about the appropriateness of the data for this research question.

First, it is true that our sample was not a random one. Individuals who got killed by humans or hit by car had probably a relatively “bold” personality and thus probably those shier, harder-to-detect wolves were not found dead, as their behavior exposed them less, or made them die in more remote places. But on the other hand, it is basically impossible to conduct any study on wolves, or wildlife in general, with random sampling. The only studies where you can obtain observations at random from a population are perhaps those based on non-invasive methods, like camera trapping. Still, even with those non-invasive methods, detectability is affected by personality traits of wildlife individuals (https://www.jstor.org/stable/3873251). From a technical viewpoint, also those studies that live-trapped wolves, like the one you cited from Mancinelli et al., did not sample individuals at random from the population. Probably, shier, and hard-to-capture individuals were systematically under sampled. Not to say that many of those studies, although interesting, trapped a low number of individuals, and even in case their extraction mechanism was “as-good-as-random”, they can hardly be taken as based on a representative sample, due to limited sample size.

Most importantly, our sample, although not randomly extracted, allowed for interpolation: the fact that we did not find an effect of anthropization over wolf growth or dimorphism is a stable truth for our sample. In case, it might be unwise to generalize out findings, by extrapolating our conclusions to an entire population which was not homogeneously covered by sampling. Or to different contexts. Which we did not do. In the Discussion we encouraged further research exactly for that reason, although probably a satisfactory sampling strategy for wolves will never be reached, at least from a statistical viewpoint. See lines XXXX.

Comment: For model selection, you noted in L195-197 that you used a backward approach and started with the most complex model, then removing one term per time. Was there a particular aim in using this method instead of a forward approach, starting with examination of the effects of single variables?

Reply: in statistics we acknowledge that there are multiple approaches to model selection, and that on some occasions, models are not even selected but simply overcontrolled (e.g., when accounting for pre-treatment differences in experiments) or averaged.

In our case, we used the back wise approach as we relied on the back door criterion. Namely, we made some clear assumptions about the sources of variability that should have been controlled for, to identify the effect of urbanization on the growth and dimorphism (see Fig. S5), given some meaningful ecological process. Then we used cross validation and the inspection of model coefficients to decide if our theoretically sound sources of variation were also supported by the data. In our opinion this approach is just as “exploratory” as the stepwise approach, but sounder: one must clearly specify potential sources of variation in advance and then test them one by one. This prevents the omission of potential sources of variation and then spurious findings in model coefficients, caused by omitted variable bias. 

 “Elwert F. Graphical causal models. In: Morgan SL, editors. Handbook of causal analysis for social research. Springer, New York; 2013. pp. 245-273.” (cited in the manuscript) gives insight on the benefits and shortcomings of this approach, which in biology is unfortunately limited mostly to structural equation models or path analysis, but which in practice is more generalizable.

Comment: L35, L152, L266 and elsewhere: please pay attention to the difference between “to grow” (“increase in size or substance” and “to grow up” (usually meaning “to reach maturity; become adult”). Given the wording here I think you mean the former. This term is quite important in your text, but its use is not quite clear.

Reply: thanks for this suggestion. We replaced “to grow up” with “to grow”.

Comment: L35-37: This phrase, stating with “Because” appears incomplete, as readers would expect “Because X, therefore Y” and the last part is missing. Overall, I think the abstract needs to be reworded and reorganized to clarify the study background, aims and conclusions (please also see the above comments).

Reply: changed as requested.

Comment: L47: Words that are in the title usually do not need to be included in the keywords and vice versa. Here, I would suggest keeping “Canis lupus” in the key words but not in the title, as there are no other wolf species in Europe, and retain “grey wolves” in the title.

Reply: changed accordingly.

Comment: L66: “in some areas”

Reply: we completely re-wrote the introduction, please check it now.

Comment: L75: please remove the comma after “Italy”

Reply: we completely re-wrote the introduction, please check it now.

Comment: L77: Altered legal status and social changes may also have been important here (see e.g., Chapron et al. 2014, cited in the text).

Reply: changed accordingly.

Comment: L78: This sentence is not clear; if I understand the meaning correctly I suggest “Considering” at the start, and removing “all the”, as “the” implies that you have already explained what these scenarios are, whereas this information follows below the statement.

Reply: we completely re-wrote the introduction, please check it now.

Comment: L79-80: Do you mean “wolves could not efficiently replace large ungulates as a food source”?

Reply: the main idea is that although wolves could replace large ungulates with smaller preys, these might not be energetically convenient, as they require comparatively more foraging and provide animals with limited nutrition. See lines XXX.

Comment: L84: It is not fully clear what “this gap” refers to

Reply: we changed this section. Please see lines XXX.

Comment: L86: “In [or “across”] a time span of 22 years”

Reply: changed accordingly.

Comment: L101: “illegal wolf killings”

Reply: changed accordingly.

Comment: L104: I think you here mean the exodus of humans?

Reply: yes, changed accordingly.

Comment: L110: Suggested “available throughout the region”

Reply: changed accordingly.

Comment: L120-132: After the section headline, I recommend starting with L125-132 that explains your sampling and then L120-124 on how you assessed each sample.

Reply: changed accordingly.

Comment: L131: Should this procedure, when carried out by humans on wolves (or other animals), not be described as “necropsy”?

Reply: changed accordingly.

Comment: L139 and throughout the text: The offspring of wolves are usually called “pups” and not “cubs” (e.g., Barber-Meyer et al. 2021, https://doi.org/10.1016/j.biocon.2021.109145) although you have used “pup/s” in L268 and L275.

Reply: changed accordingly.

Comment: L143-145: It is unclear what “rather” means in this phrase, and when you state “ensured a higher level of flexibility”, it is not obvious what this is measured against (another type of predictor?)

Reply: we wanted to emphasize that using body length as a predictor of body mass, is analogous to use the ratio between body mass and body length. But more flexible, as an approach, because it allows for non-linear or heteroskedastic associations between the two.

Comment: L147: Is this your finding, or are you referring to other studies?

Reply: in case you referred to using body length as a predictor for body mass, rather than a ratio between the two measures, we emphasized the higher flexibility that can be attained by using covariates. In facts, this approach principle also allows to account for nonlinear interactions or heteroskedasticity, a common issue in morphometry. See also the new citation that we included in the text.

Comment: L148 and throughout the text: Usually “prey” is used, also in plural.

Reply: changed accordingly. 

Comment: L161-166: I suggest rewording this section for clarity, starting with L164-166 to first explain that you are expecting increased body mass in both sexes because of anthropogenic food, and then note that you are expecting reduced growth in males because of reduced selective pressures for larger body size, followed by L163-164.

Reply: changed accordingly. See lines XXX.

Comment: 266: I am not sure what is meant with “grew up in a less pronounced way”

Reply: see lines XXX.

Comment: L270: Please remove the comma after “Perhaps”

Reply: changed accordingly. 

Comment: L284: Please check this line after “opens”

Reply: changed accordingly.

Comment: L298: “associated with”

Reply: changed accordingly.

Comment: L299: “Considering the”

Reply: changed accordingly.

Comment: L304: It is not clear what “it” refers to in this phrase. Also, for this section, L299-302 and L302-304 are not well connected, and please see the general comments above.

Reply: changed accordingly.

Reviewer #2

Comment: The manuscript deals with an interesting topic. The Authors investigated if and how anthropization can affect fitness-related traits of the gray wolf in Europe. The study is based on good material, results may be used in the population conservation of the wolf. Nevertheless, I have found some parts in the manuscript which need improvement.

Comment: Should handle statistical analyzes and hypotheses separately. A part of it fits better in the Introduction.

Reply: also following what has been suggested by Reviewer#1 we moved part of our Methods in the Introduction. See lines XXXX.

Comment: Most samples do not come from random sampling (roadkill). I recommend mentioning this.

Reply: we mentioned this point (see lines XXXX). However, see also our reply to the comment above, from Reviewer#1.

Comment: Instead of Figures S1 and S2, the sample numbers should be presented in a summary table according to age group, sex, year of collection and mortality pattern. The number of samples per year is relatively small.

Reply: We would like to maintain Appendix 1, as it shows differences between the two study areas. However, we now edited Table 1, where a short overview of our sample is provided.

Comment: From Line 119. It should be mentioned whether there were wolf-dog hybrids among the examined wolves, the seasonality of reproduction may differ from that of wolves.

Reply: Unfortunately, not all of our individuals had their DNA tested. At the time of the study (and even today). By lacking some information about multiple individuals, accounting for this issue would ultimately be impossible and thus mentioning it would also be more confusing to readers. We agree in principle that hybrids may have different biological characteristics that contributed to made them different from pure wolves but at the moment there are no firm evidence that a clear difference exists. We have mentioned the possible presence of hybrids in our sample that, being randomly sampled trough car accidents would suggest an equal occurrence of the two categories.

Comment: Based on a study, the authors considered May 1 the date of birth. There can be a difference of up to two months in reproduction and, thus, in the time of pupping. Studies should be mentioned from other seasonally breeding carnivorous species, which estimated the young's age similarly.

Reply: This decision was based on the available literature about the reproduction timing of wolves in Italy. Our approach was the only way to back-date wolves and estimate their age at the time when they had been found. 

Wolves have a palearctic distribution, and thus we totally agree that their breeding season might differ considerably across different ecosystems. But yet we deemed reasonable to make our calculations on available evidence from Italy only. Apart from our cited literature, guidelines from the Large Carnivore Group of the Autonomous Province of Trento (https://grandicarnivori.provincia.tn.it/Il-lupo/BIOLOGIA-HABITAT-E-DISTRIBUZIONE/RIPRODUZIONE) state that wolves in Italy breed mostly on early May.

We acknowledge that not knowing exactly the time of the reproduction might be a limitation of our study. And we now discussed this at lines XXXX. However, absent any other evidence, we believe our choice to be a reasonable assumption.

Comment: In the Introduction and Discussion sections, studies from other social and non-social carnivore species should be mentioned, which analyze the influence of the human environment and the shift in body weight resulting from changes in the food supply.

Reply: we understand that broadening our references would be nice, but we already mentioned what we found across literature and deemed worth citing (references 14 to 22 for various large carnivores, and 25-27 for the gray wolf.). We found relatively few studies that compared shifts in the body weight of large carnivores due to anthropization.

Please note that we did not consider studies about bears, in Europe and North America, because bears have an omnivorous diet, and are thus more flexible in their food habits than other large carnivores. Moreover, both in North America and some European countries, bears are also fed by hunters, so patterns in their body mass across areas with different human presence also depend upon deliberate foraging and not just changes in food supply from waste. Moreover, we did not consider those carnivores whose body mass varies in response to human presence because their natural preys are depleted through bushmeat, as these dynamics do not occur in Europe.

Anyway, any suggestion about further references, that might improve the Introduction and Discussion is welcome.

Comment: The Discussion sometimes lacks references or examples for findings (e.g., Lines 249-251, 257, 266-273, 297-299, 321-322.

Reply: We acknowledge that the number of references referred to similar cases is not high, but the reason is the modest amount of literature (at least, to the best of our knowledge), about the growth and sexual dimorphism of wolves, and their interplay with environmental conditions. We would gladly compare our findings with insights from other studies, especially from Europe or Italy, but we were seriously limited by the restrict number of other studies. Anyway, we would be grateful to you if you would be aware of any particular studies that might be useful and should be cited.

---

## [Decision Letter · Decision Letter 1]

30 Jan 2023

PONE-D-22-27536R1

A human-neutral large carnivore? No patterns in the body mass of gray wolves across a gradient of anthropization

PLOS ONE

Dear Dr. Cerri,

Thank you for submitting your manuscript to PLOS ONE. After careful consideration, we feel that it has merit but does not fully meet PLOS ONE’s publication criteria as it currently stands. Therefore, we invite you to submit a revised version of the manuscript that addresses the points raised during the review process.

We look forward to receiving your revised manuscript.

Kind regards,

Paulo Corti, Ph.D.

Academic Editor

PLOS ONE

Journal Requirements:

Additional Editor Comments:

Please answer the questions from reviewer #2. I addition, from the previous review of referee #1, please include the line number of your manuscript in your replay.

Reviewers' comments:

Reviewer's Responses to Questions

**Comments to the Author**

1. If the authors have adequately addressed your comments raised in a previous round of review and you feel that this manuscript is now acceptable for publication, you may indicate that here to bypass the “Comments to the Author” section, enter your conflict of interest statement in the “Confidential to Editor” section, and submit your "Accept" recommendation.

Reviewer #2: (No Response)

2. Is the manuscript technically sound, and do the data support the conclusions?

Reviewer #2: Yes

3. Has the statistical analysis been performed appropriately and rigorously? 

Reviewer #2: Yes

4. Have the authors made all data underlying the findings in their manuscript fully available?

Reviewer #2: Yes

5. Is the manuscript presented in an intelligible fashion and written in standard English?

Reviewer #2: (No Response)

6. Review Comments to the Author

Reviewer #2: The corrected manuscript contains additions at several points that I requested in the review. However, I still recommend a few additions.

Overall, the reliability of the data is not in question, but the Authors must describe the data quality as precisely as possible, excluding or mentioning influencing and limiting circumstances based on objective criteria. This is important because the Authors' first study on this topic can be a reference for further similar studies.

As the Authors write, it is well known that wolf-dog hybrids are present in a considerable proportion of the Italian wolf population. The Reader will also know this; that is, it is not the fact that this question is dealt with in the methodology that causes confusion, but that it weakens the results if it is omitted. In the Methods (Collection of dead wolves and laboratory analysis subchapter), I propose a short addition in which it is described that during the protocol used in the collection of the carcass and the autopsy, based on morphological and other phenotypic characteristics, hybrid-suspicious individuals were excluded. Thus, hybrids did not occur, or in the absence of DNA analysis, they may have been included in the analysis in a small proportion. (If this statement is true.) Please add, even with a new reference, how the taxonomic classification of the dead individuals took place.

A higher proportion of hybrids in the examined sample would be problematic.

In the Discussion section, the authors need to know what they meant by those statements that lacked a reference.

For example:

Lines 269-271 (corrected manuscript). Changes in food sources and the relationship between food sources and body condition were not studied. Any statements related to this topic must be referred to.

Line 277. No other large carnivores than grey wolf were studied; therefore should be mentioned species or rather references here.

Lines 291-292. Given that no food source analysis was performed, I also recommend citing the literature supporting the statement for this sentence, "This suggests a positive influence of human-derived food sources...".

Lines 318320. The relationship between body condition and reproduction was also not included in the analysis of the study, so please cite here the publication based on which this sentence was written.

For the topic of body weight change, further studies on canids could have been considered. (The omnivorous brown bear and mustelids are inappropriate in this respect.) The question has been answered.

7. PLOS authors have the option to publish the peer review history of their article (what does this mean?). If published, this will include your full peer review and any attached files.

Reviewer #2: No

---

## [Author Response · Author response to Decision Letter 1]

7 Feb 2023

################################################################################

PONE-D-22-27536R1 - A human-neutral large carnivore? No patterns in the body mass of gray wolves across a gradient of anthropization

################################################################################

Dear Dr. Cerri,

Thank you for submitting your manuscript to PLOS ONE. After careful consideration, we feel that it has merit but does not fully meet PLOS ONE’s publication criteria as it currently stands. Therefore, we invite you to submit a revised version of the manuscript that addresses the points raised during the review process.

We look forward to receiving your revised manuscript.

Kind regards,

Paulo Corti, Ph.D.

Academic Editor

PLOS ONE

################################################################################

Journal Requirements:

################################################################################

################################################################################

Additional Editor Comments:

################################################################################

Comment: Please answer the questions from reviewer #2. I addition, from the previous review of referee #1, please include the line number of your manuscript in your replay.

Reply: we replied in detail to each comment raised by Reviewer #2 and we added line counts, as requested by Reviewer#1.

################################################################################

Reviewers' comments:

################################################################################

1. If the authors have adequately addressed your comments raised in a previous round of review and you feel that this manuscript is now acceptable for publication, you may indicate that here to bypass the “Comments to the Author” section, enter your conflict of interest statement in the “Confidential to Editor” section, and submit your "Accept" recommendation.

Reviewer #2: (No Response)

################################################################################

2. Is the manuscript technically sound, and do the data support the conclusions? - The manuscript must describe a technically sound piece of scientific research with data that supports the conclusions. Experiments must have been conducted rigorously, with appropriate controls, replication, and sample sizes. The conclusions must be drawn appropriately based on the data presented.

Reviewer #2: Yes

################################################################################

3. Has the statistical analysis been performed appropriately and rigorously?

Reviewer #2: Yes

################################################################################

4. Have the authors made all data underlying the findings in their manuscript fully available? - The PLOS Data policy requires authors to make all data underlying the findings described in their manuscript fully available without restriction, with rare exception (please refer to the Data Availability Statement in the manuscript PDF file). The data should be provided as part of the manuscript or its supporting information, or deposited to a public repository. For example, in addition to summary statistics, the data points behind means, medians and variance measures should be available. If there are restrictions on publicly sharing data—e.g. participant privacy or use of data from a third party—those must be specified.

Reviewer #2: Yes

################################################################################

5. Is the manuscript presented in an intelligible fashion and written in standard English? - PLOS ONE does not copyedit accepted manuscripts, so the language in submitted articles must be clear, correct, and unambiguous. Any typographical or grammatical errors should be corrected at revision, so please note any specific errors here.

Reviewer #2: (No Response)

################################################################################

6. Review Comments to the Author - Please use the space provided to explain your answers to the questions above. You may also include additional comments for the author, including concerns about dual publication, research ethics, or publication ethics. (Please upload your review as an attachment if it exceeds 20,000 characters)

Comment: Reviewer #2: The corrected manuscript contains additions at several points that I requested in the review. However, I still recommend a few additions. Overall, the reliability of the data is not in question, but the Authors must describe the data quality as precisely as possible, excluding or mentioning influencing and limiting circumstances based on objective criteria. This is important because the Authors' first study on this topic can be a reference for further similar studies. 

Reply: we addressed each point you raised in detail, and we now hope to have described data quality as precisely as possible. Please, see lines 157-183.

Comment: As the Authors write, it is well known that wolf-dog hybrids are present in a considerable proportion of the Italian wolf population. The Reader will also know this; that is, it is not the fact that this question is dealt with in the methodology that causes confusion, but that it weakens the results if it is omitted. In the Methods (Collection of dead wolves and laboratory analysis subchapter), I propose a short addition in which it is described that during the protocol used in the collection of the carcass and the autopsy, based on morphological and other phenotypic characteristics, hybrid-suspicious individuals were excluded. Thus, hybrids did not occur, or in the absence of DNA analysis, they may have been included in the analysis in a small proportion. (If this statement is true.) Please add, even with a new reference, how the taxonomic classification of the dead individuals took place. A higher proportion of hybrids in the examined sample would be problematic.

Reply: Our sample did not include a large proportion of hybrids as out of 185 individuals only 11 were not tested genetically and in the other 174 only 8 resulted recent hybrids (F1 or F2), so we have not excluded these individuals from analyses. 

Although we acknowledge that this is a limit of our research, and that future studies should explore the extent to which hybridization could affect wolf behavior and fitness in anthropized environments, we did not discard hybrid individuals, because evidence for potential differences is still weak and largely unknown.

While during domestication dogs started producing alpha-amylase (https://doi.org/10.1038/nature11837), amylase activity is highly variable between dog breeds and even between individuals of the same breed, due to a strong heterogeneity in the number of copies of the gene AMY2B (https://doi.org/10.1111/age.12179). Therefore, provided that for wolves we do not know if AMY2B is associated to amylase activity just like in domestic dogs, hybrid wolves could have simply originated from dogs with a low number of copies of AMY2B, and therefore they could have a low amylase activity and no increased capacity to exploit starch-rich human food waste at all.

As for other behavioral traits, associated to increased boldness towards humans, which dogs developed during domestication, our current understanding of the underlying genetic mechanisms is still limited (e.g., https://www.frontiersin.org/articles/10.3389/fvets.2021.693290/full). Also considered that wolves in Italy were also subjected to long-term pressures from prolonged coexistence with humans, a phenomenon which could have masked any effect of hybridization on the selection of “bolder” individuals.

While we acknowledge that these potential mechanisms provide fascinating directions for future studies, and certainly deserve further attention (like we mentioned in the Discussion), we do not deem them to constitute a solid basis of evidence for discarding non-tested wolves from our datasets.

We now reported the number of wolves who were tested for hybridization, as well as the number of recent hybrids in our sample. Moreover, we provided a short description on how we carried out the classification of the dead individuals. Please see lines 168-183.

Comment: Lines 269-271 (corrected manuscript). Changes in food sources and the relationship between food sources and body condition were not studied. Any statements related to this topic must be referred to.

Reply: we understand that it is important to contextualize this point. We now emphasized, that our analyses dealt with landscape-scales responses and not with fixed or time-variable resource selection. Please see lines 275-277.

Comment: Line 277. No other large carnivores than grey wolf were studied; therefore should be mentioned species or rather references here.

Reply: we now shortly explained the content of the two references. Please see lines 283-288.

Comment: Lines 291-292. Given that no food source analysis was performed, I also recommend citing the literature supporting the statement for this sentence, "This suggests a positive influence of human-derived food sources...".

Reply: Unfortunately we found no references that we could cite in support of what we suggested as an hypothesis. There are very few studies exploring the diet of wolf pups in Europe, let alone the diet of wolf pups in anthropized areas. Here we are just hypothesizing a mechanism (we used “this suggests...”) to explain the pattern that we observed, which is plausible considered the that pups depend upon adults in their first months.

Comment: Lines 318-320. The relationship between body condition and reproduction was also not included in the analysis of the study, so please cite here the publication based on which this sentence was written.

Reply: we now added the reference as requested. See lines 332 and reference n.96.

Comment: For the topic of body weight change, further studies on canids could have been considered. (The omnivorous brown bear and mustelids are inappropriate in this respect.) The question has been answered.

Reply: We mentioned studies about the effect of anthropogenic food subsidies that dealt with a range of different species, other than wolves, because there is relatively little research on this topic. In the study we tried to focus on potential responses of top predators, what gray wolves are, to anthropization, also by considering the potential exploitation of anthropogenic food sources.

################################################################################

7. PLOS authors have the option to publish the peer review history of their article (what does this mean?). If published, this will include your full peer review and any attached files. - Do you want your identity to be public for this peer review? For information about this choice, including consent withdrawal, please see our Privacy Policy.

Reviewer #2: No

---

## [Editor Report · Decision Letter 2]

13 Feb 2023

A human-neutral large carnivore? No patterns in the body mass of gray wolves across a gradient of anthropization

PONE-D-22-27536R2

Dear Dr. Cerri,

We’re pleased to inform you that your manuscript has been judged scientifically suitable for publication and will be formally accepted for publication once it meets all outstanding technical requirements.

Kind regards,

Paulo Corti, Ph.D.

Academic Editor

PLOS ONE

---

## [Editor Report · Acceptance letter]

17 Feb 2023

PONE-D-22-27536R2 

A human-neutral large carnivore? No patterns in the body mass of gray wolves across a gradient of anthropization 

Dear Dr. Cerri:

I'm pleased to inform you that your manuscript has been deemed suitable for publication in PLOS ONE. Congratulations! Your manuscript is now with our production department. 

Kind regards, 

on behalf of

Dr. Paulo Corti 

Academic Editor

PLOS ONE